# GNNUpdater: Adaptive Self-Triggered Training Framework on Dynamic Graphs

## Abstract

Adapting Graph Neural Networks (GNNs) to evolving, dynamic graph data poses a difficult operational problem: when should these models be updated? At scale, retraining is expensive, so updates should be triggered only when the expected performance gain justifies the GPU cost. This problem is especially challenging in graph settings because of two factors: *label delay*, where ground truth arrives long after predictions are made, and *hidden drift*, where structural dependencies propagate changes through multiple hops and degrade performance unexpectedly. We propose **GNNUpdater**, an adaptive framework that decides when to trigger GNN training. It addresses these challenges with two components: (1) a *performance predictor* that estimates model quality from node embedding shifts, removing dependence on immediate ground-truth labels, and (2) a *graph-aware update trigger* that uses label propagation to detect widespread performance degradation across the graph. We implement GNNUpdater as a distributed streaming-GNN library for billion-edge dynamic graphs. Experiments show that GNNUpdater outperforms periodic, performance-based, and drift-detection baselines at comparable GPU-hour budgets, or matches their performance with substantially less training cost. The implementation can be found at the anonymous link: `https://anonymous.4open.science/r/GNNUpdater-B47D/`.

## 1 Introduction

In machine learning systems deployed in dynamic environments, models inevitably grow stale. Shifting data distributions, or *concept drift* Gama et al. (2014); Lu et al. (2018); Webb et al. (2016), silently degrade predictive performance over time. The central operational question is therefore: *when should we update the model?* This is fundamentally an *economic decision*—especially at large scale: retraining models on continuously arriving, large-scale data consumes enormous computational resources—frequently hundreds or thousands of GPU-hours per cycle—representing substantial infrastructure and energy cost. Updating too frequently wastes budget on marginal gains, while updating too slowly incurs prolonged periods of degraded predictions that may violate the application's *Service Level Objective (SLO)*. Conventional approaches typically rely on periodic retraining Azure (2025); Cloud (2025), performance-based triggers Nigenda et al. (2022); Bifet & Gavalda (2007), or statistical distribution monitoring Lu et al. (2018); Gama et al. (2004).

This trade-off becomes particularly acute for Graph Neural Networks (GNNs) operating on continuously evolving graphs—such as social networks, financial transaction streams, and e-commerce platforms Kumar et al. (2019); Wang et al. (2021a); Lin et al. (2022); Fan et al. (2019); Borisyuk et al. (2024); Weber et al. (2019)—characterized by the continuous arrival of new nodes, edges, and evolving node/edge features. Retraining GNNs on large-scale dynamic graphs with millions of nodes and edges is extremely expensive, often consuming hundreds of GPU-hours and massive energy per update cycle Lin et al. (2022); Shao et al. (2024). While the problem of optimal update timing is well-recognized in traditional ML NannyML (2023); Evidently (2023); Van Looveren et al. (2019); Dong et al. (2024); Lyu et al. (2024), it remains significantly underexplored for graph-structured data Yuan et al. (2023). As GNNs are increasingly deployed in many critical real-world applications—such as recommendation systems Wu et al. (2022); Ying et al. (2018), fraud detection Liu et al. (2021b); Weber et al. (2019), and traffic forecasting Chen et al. (2021); Jiang et al. (2023)—determining *when* to trigger updates in streaming graph settings has become a vital yet underaddressed challenge with direct implications for system reliability and operational cost.

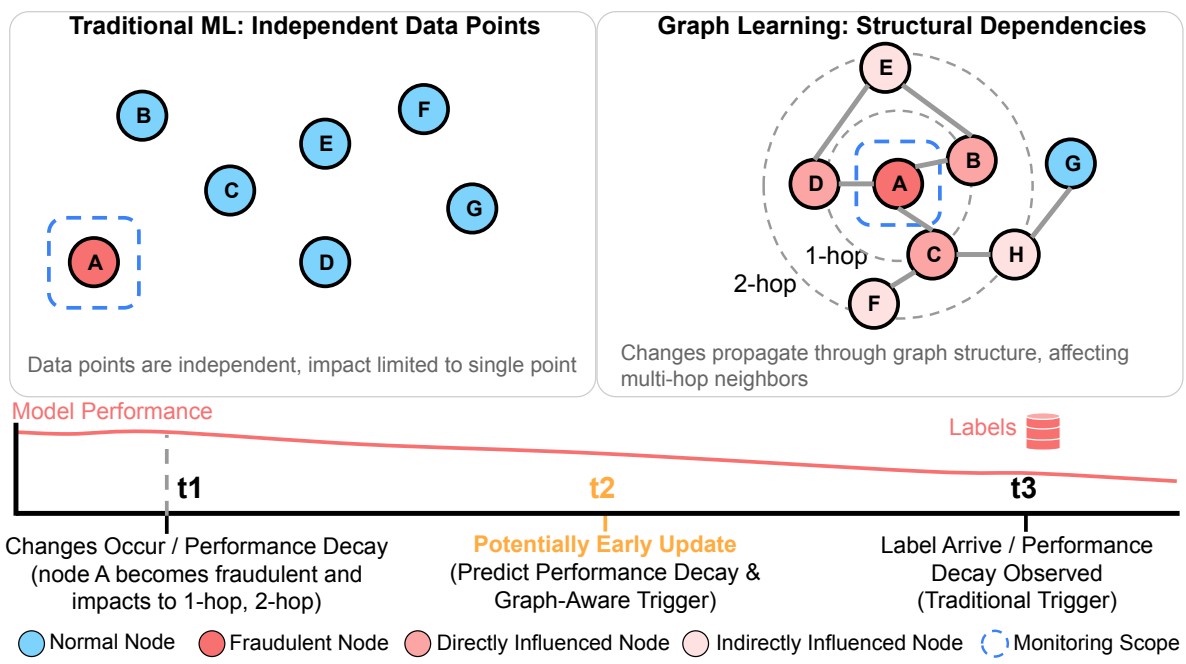

Figure 1: Challenges of GNN update timing. *Top*: In traditional ML (left), data points are independent. In graph learning (right), a change to one node propagates through its multi-hop neighborhood, causing *hidden drift*. *Bottom*: Performance decays at $t_1$, but labels only arrive at $t_3$ (label delay). A traditional trigger updates too late. Our graph-aware trigger monitors embedding shifts to predict performance decay, enabling an early update at $t_2$.

**Key Challenges.** Deciding the optimal update trigger for GNNs is inherently more complex than in traditional ML due to two primary obstacles (Fig. 1):

**(1) Label Delay.** Standard performance-driven triggers Nigenda et al. (2022) rely on the immediate availability of ground truth to detect accuracy drops. However, in many graph-centric domains, labels are delayed by design: fraud verification can take weeks Wang et al. (2021a), user feedback in recommendation engines matures over several days Ni et al. (2019); Ktena et al. (2019), and credit default indicators may only surface months post-origination Wang et al. (2023). By the time a performance dip is captured through reactive monitoring, the stale model may have already generated erroneous predictions for a prolonged duration.

**(2) Hidden Drift.** Unlike traditional ML where data points are independent, nodes in a graph are structurally interdependent. GNNs generate node embeddings by recursively aggregating features from a node's $k$-hop neighborhood Kipf & Welling (2017); Hamilton et al. (2017); Veličković et al. (2018). This dependency creates a ripple effect: a localized change (e.g., a small cluster of fraudulent accounts linked to legitimate users) propagates through the graph topology during message passing. As a result, embeddings of distant, inactive nodes can be silently altered—even when those nodes themselves have received no new incident edges or feature updates. By the time these inactive nodes are queried in downstream tasks, their predictions degrade sharply. This turns a localized perturbation into widespread, systemic prediction failure. Traditional triggers fail here because they monitor only recently active nodes (i.e., those with incident edges or feature updates in the current batch), creating a critical blind spot for this latent, propagating degradation, which we term *hidden drift*.

**Our Approach.** To overcome these challenges, we introduce GNNUpdater, a model-agnostic framework that treats GNN retraining as a cost-accuracy optimization problem. Operating as a strategic wrapper around deployed models, it employs two synergistic mechanisms:

**Performance Predictor** (§3.2) bypasses label delay by utilizing *embedding shift* as a high-fidelity proxy for model health. We maintain a set of reference embeddings from the most recent update and monitor their divergence from live embeddings generated during inference. Significant drift indicates that the model is navigating out-of-distribution regions. We aggregate these shifts across the neighborhood of each node (mirroring the GNN's local aggregation

logic) and employ a lightweight regressor to map these drift signatures to predicted performance metrics. The resulting performance predictions achieve a Pearson correlation of up to 0.96 with actual performance, enabling proactive detection without requiring immediate ground-truth labels.

**Graph-Aware Update Trigger** (§3.3) mitigates hidden drift by diffusing degradation signals throughout the graph manifold. For every incoming batch, we categorize active nodes as either *problematic* (whose predicted performance, as estimated by our Performance Predictor (§3.2), falls below the application-defined SLO $\varepsilon$) or *normal*. We then utilize label propagation Zhu & Ghahramani (2002) to structurally align our detection mechanism with the GNN's own error propagation logic, diffusing these status labels across the graph topology. This process uncovers latent degradation in inactive or unmonitored neighborhoods. An update is only initiated when the global density of problematic nodes exceeds a threshold $\phi$, signaling *systemic* rather than isolated failure. Crucially, $\phi$ serves as a *cost-control lever*: higher values prioritize budget conservation by deferring updates, while lower values prioritize model precision, allowing operators to calibrate the accuracy-cost trade-off according to specific economic constraints.

**Contributions.**

- We formulate the GNN update timing problem as an *economic optimization* that explicitly balances prediction loss against training cost, and identify *label delay* and *hidden drift* as fundamental challenges distinct from traditional ML (§ 2).

- We propose a *label-delay tolerant performance predictor* using neighborhood-aggregated embedding shifts, achieving up to 0.96 Pearson correlation with true performance (§3.2).

- We design a *graph-aware update trigger* using label propagation to detect *systemic degradation* across the entire graph, enabling early detection of multi-hop drift (§3.3).

- We implement GNNUpdater as an efficient distributed system with block-based graph storage and GPU-accelerated sampling, reducing graph operation overhead by up to *92.5%* compared to DGL (§3.4).

- Through extensive experiments on billion-edge graphs, we show GNNUpdater improves accuracy by *5.3% on average* (*up to 34%*) at an equal GPU budget, or reduces training cost by *2.0× on average* (*up to 6×*) at matched performance (§5).

## 2 Preliminary and Problem Formulation

Table 1: Key Notations.

| Symbol | Description |
|---|---|
| $G_t, \Delta G_t$ | Cumulative graph and batch at time $t$ |
| $\mathcal{T}_t, \mathcal{Y}_t$ | Target nodes and labels at time $t$ |
| $\tau, W$ | Label delay and retraining window size |
| $\theta_t, \delta_t$ | Model parameters and update decision at time $t$ |
| $C_{\text{train},t}, \lambda$ | Training cost and accuracy-cost exchange rate |
| $\hat{P}_{v,t}$ | Predicted performance for node $v$ at time $t$ |
| $\mathbf{H}_t, \mathbf{h}_v^t$ | Global and node-level embeddings at time $t$ |
| $\mathbf{h}_v^{\text{ref}}$ | Reference embedding for node $v$ |
| $\text{drift}(v)$ | Aggregated embedding drift score for node $v$ |
| $\mathcal{R}_t$ | Global degradation ratio at time $t$ |
| *Operational Knobs* | |
| $\varepsilon$ | Performance SLO |
| $\phi$ | Systemic failure trigger threshold |

**Streaming Graphs.** We observe a sequence of temporal edge-update batches $\{\Delta G_t\}_{t=1}^T$, where $\Delta G_t = \{(u_i, v_i, t_i)\}_{i=1}^{n_t}$ consists of $n_t$ edges arriving in window $t$. Each node $v$ has a feature vector. The cumulative graph is $G_t = (V_t, E_t)$

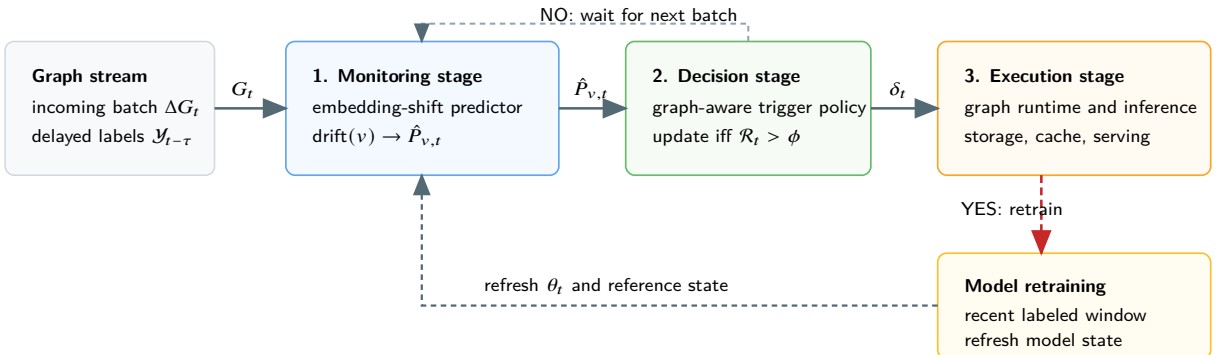

Figure 2: The overview of GNNUpdater. The framework operates as a closed-loop pipeline comprising three logical stages: (1) Monitoring Stage (§3.2), where the Performance Predictor estimates model degradation via embedding shifts; (2) Decision Stage (§3.3), where the Graph-Aware Trigger determines the optimal retraining timing; and (3) Execution Stage (§3.4), where the system performs efficient graph updates and model retraining.

with $V_t = \bigcup_{i \leq t} V(\Delta G_i)$ and $E_t = \bigcup_{i \leq t} E(\Delta G_i)$. A subset $\mathcal{T}_t \subseteq V_t$ consists of *target nodes*: each $v \in \mathcal{T}_t$ has a true label $y_v$, which is generated in window $t$ but arrives after a $\tau$-step delay. We denote by $\mathcal{Y}_t = \{ y_v | v \in \mathcal{T}_t \}$ the set of labels associated with batch $t$.

**Concept Drift in Dynamic Graphs.** The evolving nature of streaming graphs gives rise to several types of concept drift, which can degrade model performance Lu et al. (2018). Let $x$ denote the input graph data (including topology and node features), and $y$ denote the target labels. We identify two primary categories: (1) *Structural and Feature Drift* (a form of $p(x)$ drift), which includes changes to the graph topology or node features, and (2) *Semantic Drift* (a form of $p(y|x)$ drift), where the underlying relationship between graph features and target labels changes.

**GNN Inference and Training on Streaming Graphs.** We maintain a GNN with parameters $\theta$. At each time $t$, we (i) infer predictions $\hat{y}_v$ for $v \in \mathcal{T}_t$ on $G_t$ using $\theta_{t-1}$ (a forward pass of the GNN), and (ii) optionally update via a binary decision $\delta_t \in \{0, 1\}$. If $\delta_t = 1$, we fine-tune on the most recent $W$ steps of available labels:

$$\theta_t = \text{Train}\big(\theta_{t-1}, G_t, \mathcal{Y}_{[t-\tau-W+1:t-\tau]}\big),$$

and denote its training GPU cost by $C_{\text{train},t}$. Otherwise, $\theta_t = \theta_{t-1}$.

**Update Timing Problem.** Let $\ell(\theta_t; G_t, \mathcal{Y}_t)$ denote the task loss (e.g., cross-entropy) computed on the true labels for batch $t$, which become available only after a delay $\tau$. Given an initial model parameter $\theta_0$, the update timing problem aims to solve the following optimization problem:

$$\min_{\{\delta_t\}_{t=1}^T} \sum_{t=1}^T \ell(\theta_t; G_t, \mathcal{Y}_t) + \lambda \sum_{t=1}^T \delta_t C_{\text{train},t}$$

$$\text{s.t. } \theta_t = \begin{cases} \theta_{t-1}, & \delta_t = 0 \\ \text{Train}(\theta_{t-1}, G_t, \mathcal{Y}_{[t-\tau-W+1:t-\tau]}), & \delta_t = 1 \end{cases} \tag{1}$$

where $\lambda > 0$ represents the operational exchange rate between model accuracy and computational resources (e.g., the value of a 1% accuracy gain vs. GPU hours). This formulation explicitly frames the update decision as a cost-benefit analysis: an update ($\delta_t = 1$) is only optimal if the reduction in future loss exceeds the immediate training GPU cost $C_{\text{train},t}$. Solving this requires predicting the task loss $\ell$ without access to the delayed labels $\mathcal{Y}_t$. The core of our approach is to use signals derived from the GNN's output embeddings as a proxy for this unobserved loss (detailed in §3.2), which then drive our graph-aware update decisions (§3.3).

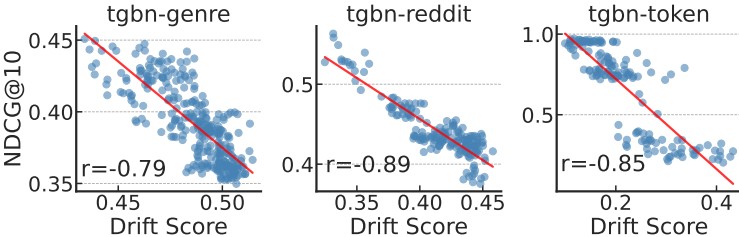
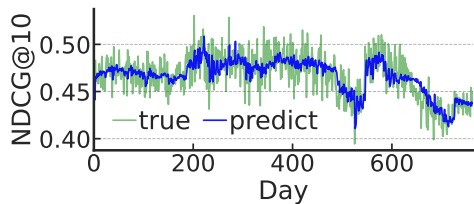

Figure 3: The embedding drift score shows a strong negative correlation with model performance across datasets for TGAT Xu et al. (2020a), with a Pearson correlation ($r$) of up to -0.89. The red line represents the linear regression fit.

Figure 4: The predicted performance closely matches the true performance, even when performance suddenly increases (caused by model updates).

# 3 The GNNUpdater Framework

GNNUpdater is an adaptive framework designed to optimize the timing of GNN retraining on dynamic graphs. By acting as a model-agnostic wrapper around a deployed GNN, it decides when to trigger an update, balancing model freshness against training GPU hours.

## 3.1 System Overview

Figure 2 summarizes the closed loop. At each window, GNNUpdater ingests the new graph batch, runs inference on the current graph, and reuses the resulting embeddings as a signal of model freshness. The monitoring stage converts these embedding shifts into node-level performance estimates without waiting for delayed labels. The decision stage aggregates those estimates over the graph and produces a binary update decision, so retraining is invoked only when degradation is widespread enough to justify its cost. If an update is triggered, the execution stage trains on the latest available labeled window and refreshes the deployed model state; otherwise, GNNUpdater keeps serving with the current model and waits for the next batch.

The rest of this section follows that pipeline: the next subsections cover performance estimation under label delay, graph-level update triggering, and the system components that support efficient updates.

## 3.2 Performance Predictor via Embedding Shift

To address the challenge of predicting model performance before ground-truth labels are available, our goal is to learn a mapping

$$f : \mathbb{R}^k \to \mathbb{R},$$

$$f(\mathbf{x}_{v,t}) = \hat{P}_{v,t} \approx \text{model performance for node } v \text{ at time } t \tag{2}$$

by primarily leveraging shifts in node embeddings alongside other graph-aware features. This task is approached in two main stages:

**(1) Measuring Embedding Shifts (handle $p(x)$ drift).** Here, $p(x)$ refers to the input graph data distribution (topology and features). We maintain two sets of embeddings for all relevant nodes: (i) *Reference embeddings* $\mathbf{H}_{\text{ref}}$, representing the node states captured by a full-graph inference performed immediately after the last model update. These serve as a stable baseline. (ii) *Current embeddings* $\mathbf{H}_t$, which are updated as new data batches arrive. Crucially, GNNUpdater is designed to be highly efficient by reusing these embeddings directly from the application's real-time inference pipeline. We denote a specific node $v$'s current and reference embedding as $\mathbf{h}_v^t \in \mathbf{H}_t$ and $\mathbf{h}_v^{\text{ref}} \in \mathbf{H}_{\text{ref}}$.[1]

We initially considered metrics such as local drift ($\|\mathbf{h}_v^t - \mathbf{h}_v^{\text{ref}}\|_1$) and global drift (MMD). Empirically, local drift is too sensitive to noise, while global drift is too slow to react. Since GNNs derive representations from multi-hop

---

[1]If node $v$ is a new node from $\Delta G_t$ and does not have its reference node embedding $\mathbf{h}_v^{\text{ref}}$, we can omit it during the computation, or use a dummy reference node embedding (e.g., all zeros).

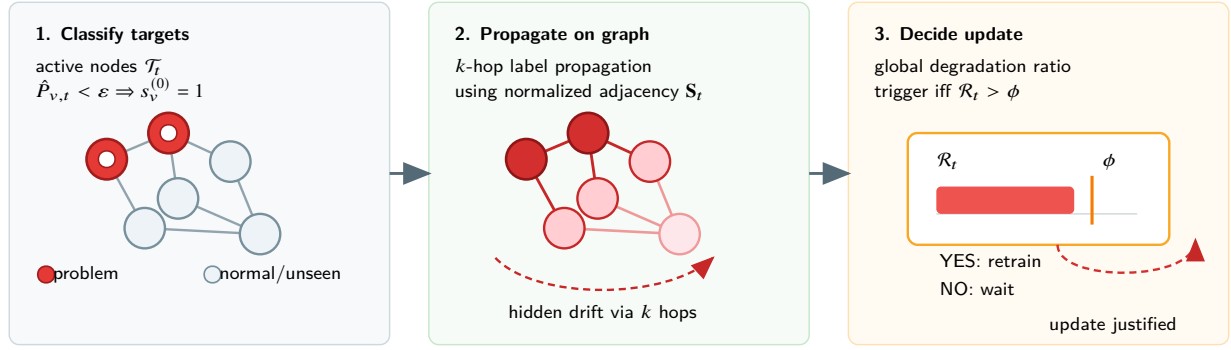

Figure 5: Overview of the Graph-Aware Trigger Policy. (Step 1) Active nodes are classified as problematic if their predicted performance $\hat{P}_{v,t}$ falls below SLO $\varepsilon$. (Step 2) Statuses are diffused via Label Propagation to capture structurally propagated drift. (Step 3) An update is triggered if the global degradation ratio $\mathcal{R}_t$ exceeds threshold $\phi$.

neighborhoods, we use an aggregated *drift score* that averages the drift of node $v$ and its immediate neighbors $\mathcal{N}(v)$:

$$\text{drift}(v) = \frac{1}{|\{v\} \cup \mathcal{N}(v)|} \left( \|\mathbf{h}_v^t - \mathbf{h}_v^{\text{ref}}\|_1 + \sum_{u \in \mathcal{N}(v)} \|\mathbf{h}_u^t - \mathbf{h}_u^{\text{ref}}\|_1 \right). \tag{3}$$

As shown in Figure 3, this score has a strong negative Pearson correlation (averaging –0.80) with downstream performance and outperforms local (–0.60) or global (–0.40) metrics.

**(2) Learning the Mapping (handle $p(y|x)$ drift).** Here, $p(y|x)$ refers to the conditional distribution of labels (semantic drift). To predict performance, we construct a feature vector $\mathbf{x}_{v,t} \in \mathbb{R}^k$ for each target node $v$ at time $t$:

$$\mathbf{x}_{v,t} = \big[ \text{drift}(v), \log(\text{num\_nodes}), \log(\text{num\_edges}), t, \deg(v) \big]. \tag{4}$$

We then collect training pairs $(\mathbf{x}_{v,t}, P_{v,t})$, where $P_{v,t}$ is the actual realized performance obtained once labels are available. We found that a Random Forest regressor Breiman (2001) yields the best results (Table 3), effectively modeling complex non-linear interactions. The regressor is periodically retrained (a fast process taking $\approx 0.25$ seconds) as new labeled data accumulates, ensuring the predictor adapts to evolving concept drift ($p(y|x)$) and maintains accuracy even if reference embeddings $\mathbf{H}_{\text{ref}}$ become stale.

### 3.3 Graph-Aware Trigger Policy

The key idea of our graph-aware update trigger is to *extrapolate local performance estimates to the entire graph*, detecting "Hidden Drift" in regions not currently being monitored.

At each time $t$, for each target node $v \in \mathcal{T}_t$, we use our predictor to estimate performance $\hat{P}_{v,t}$ and classify it as problematic (violating SLO $\varepsilon$, i.e., $\hat{P}_{v,t} < \varepsilon$) or normal ($\hat{P}_{v,t} \geq \varepsilon$). We then employ label propagation Zhu & Ghahramani (2002) to diffuse these statuses through the graph structure. We initialize a status vector $\mathbf{s}^{(0)} \in [0,1]^{|V_t|}$, where $s_v^{(0)} = 1$ if $v$ is predicted as problematic, and $s_v^{(0)} = 0$ otherwise (including unlabeled non-target nodes). Using the normalized adjacency matrix $\mathbf{S}_t = \mathbf{D}_t^{-1/2} \mathbf{A}_t \mathbf{D}_t^{-1/2}$, we perform $k$ iterations:

$$\mathbf{s}^{(\ell+1)} = \alpha \, \mathbf{S}_t \, \mathbf{s}^{(\ell)} \; + \; (1-\alpha) \, \mathbf{s}^{(0)}, \quad \ell = 0, \ldots, k-1.$$

This produces a global degradation ratio $\mathcal{R}_t = \frac{1}{|V_t|} \sum_{v \in V_t} \mathbb{I}\big[s_v^{(k)} > 0.5\big]$. If $\mathcal{R}_t > \phi$, we trigger an update. This threshold matches the economic objective in Eq. 1: isolated drifts may not justify the fixed retraining cost ($C_{\text{train},t}$), but a widespread degradation ($\mathcal{R}_t > \phi$) signals a system-wide failure where retraining is worthwhile.

Algorithm 1 summarizes this per-batch decision procedure.

**Structural Alignment with GNNs.** Label Propagation leverages the shared neighborhood aggregation operator ($S_t$) used in GNNs, providing a computationally efficient approximation to detect multi-hop drift patterns without requiring

---

**Algorithm 1** Graph-Aware Update Trigger

---

**Require:** Current graph $G_{t-1} = (V_{t-1}, E_{t-1})$, incoming edge-update batch $\Delta G_t$
**Require:** Target nodes $\mathcal{T}_t$, performance SLO $\varepsilon$, trigger threshold $\phi$
**Require:** Propagation weight $\alpha \in (0, 1)$ and propagation iterations $k$
**Ensure:** Update decision $\delta_t \in \{0, 1\}$
  1: $G_t \leftarrow G_{t-1} \cup \Delta G_t$
  2: Initialize $\mathbf{s}^{(0)} \in [0, 1]^{|V_t|}$ to all zeros
  3: $\hat{\mathbf{P}}_t \leftarrow \textsc{PredictPerf}(\mathcal{T}_t)$
  4: **for all** $v \in \mathcal{T}_t$ **do**
  5:      $s_v^{(0)} \leftarrow \mathbb{I}[\hat{P}_{v,t} < \varepsilon]$
  6: **end for**
  7: $\mathbf{S}_t \leftarrow \mathbf{D}_t^{-1/2} \mathbf{A}_t \mathbf{D}_t^{-1/2}$
  8: $\mathbf{s} \leftarrow \mathbf{s}^{(0)}$
  9: **for** $i = 1, \ldots, k$ **do**
10:      $\mathbf{s} \leftarrow \alpha \mathbf{S}_t \mathbf{s} + (1 - \alpha) \mathbf{s}^{(0)}$
11: **end for**
12: $\mathcal{R}_t \leftarrow |\{v \in V_t : s_v > 0.5\}| / |V_t|$
13: $\delta_t \leftarrow \mathbb{I}[\mathcal{R}_t > \phi]$
14: **return** $\delta_t$

---

learnable parameters or backpropagation. Specifically, standard GNNs update embeddings via $H^{(l+1)} = \sigma(S_t H^{(l)} W)$. By utilizing this operator $S_t$ to diffuse statuses, LP acts as a linear approximation of the GNN's receptive field: if a neighbor is drifting, its status propagates to $v$ via $S_t$ just as its degraded embedding would in the GNN.

**Parameters as Operational Knobs.** The parameters $\varepsilon$ and $\phi$ serve distinct operational roles. $\varepsilon$ is an application-defined SLO (e.g., min accuracy), while $\phi$ acts as a direct financial lever, allowing operators to define the specific "tipping point" where the cost of performance degradation outweighs the computational investment of retraining. Our sensitivity analysis (§5.2) shows GNNUpdater is robust to a wide range of settings.

### 3.4 Efficient System Implementation

GNNUpdater is a distributed library built on PyTorch and DGL for efficient learning on billion-edge dynamic graphs. To address the limitations of static-graph frameworks like DGL, which require expensive reconstruction during updates, GNNUpdater introduces three optimizations:

**Block-Based Streaming Graph Storage.** We implement a specialized C++ block-based storage structure (Figure 6). Each node's outgoing edges are stored as a doubly linked list of chronologically ordered lightweight blocks (typically 64-256 edges). Edge data (neighbor IDs, timestamps) resides in page-locked host memory, while lightweight metadata (pointers, timestamps) resides in GPU memory. This enables $O(1)$ edge appends and efficient temporal queries without full graph reconstruction.

**GPU-Based Neighbor Finder.** We accelerate neighbor sampling by leveraging Unified Virtual Addressing (UVA). Our custom CUDA kernels launch one thread per neighbor sample, accessing edge data directly from host memory via the GPU-resident metadata. This hybrid approach supports graphs far larger than GPU memory (billion-scale) while maintaining high throughput.

**GPU Feature Cache and Distributed Training.** GNNUpdater employs a dynamic LRU cache in GPU memory for hot node/edge features, reducing retrieval latency by 58% compared to CPU-only access. For billion-edge scale, it supports distributed training via graph partitioning and PyTorch's asynchronous RPC framework, as detailed in Appendix B.

## 4 Related Works

**Continual Graph Learning.** Continual graph learning (CGL) aims to learn incrementally on evolving graphs without catastrophic forgetting Xu et al. (2020b); Wang et al. (2021b); Zhang et al. (2024); Wang et al. (2022; 2020); Perini et al.

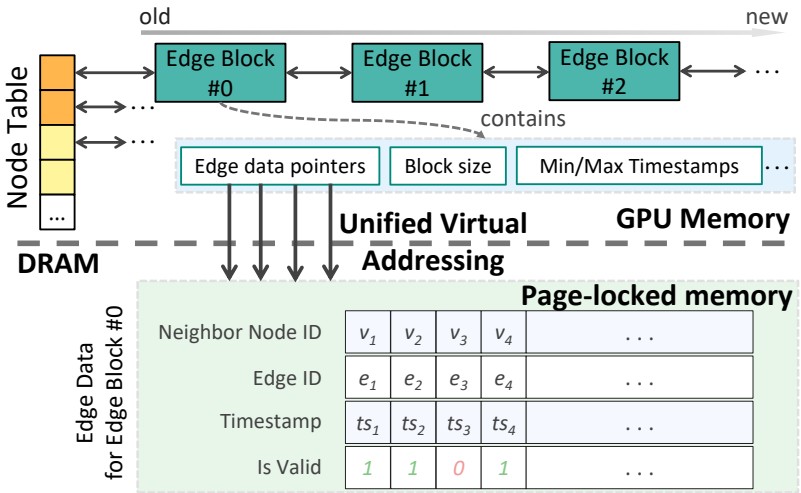

Figure 6: Block-based streaming graph storage. Each node maintains a chronological linked list of edge blocks, enabling new temporal edges to be appended without rebuilding the whole graph. Edge records remain in page-locked host memory, while compact block metadata is cached on the GPU for fast temporal neighbor lookup.

(2022); Ahrabian et al. (2021); Ding et al. (2022); Yuan et al. (2023); Su et al. (2023); De Lange et al. (2021); Parisi et al. (2019). Existing works primarily focus on mitigating forgetting when updating models Yuan et al. (2023), using techniques like parameter importance scoring Liu et al. (2021a); Kirkpatrick et al. (2017), knowledge distillation Xu et al. (2020b), experience replay Wang et al. (2020); Zhou & Cao (2021), or regularization Su et al. (2023). Unlike these works, which focus on *how* to update to preserve knowledge, GNNUpdater addresses the under-explored problem of *when* to update to balance freshness and cost.

**Learning on Dynamic Graphs.** A significant body of work focuses on developing GNN architectures for dynamic graphs Trivedi et al. (2019); Pareja et al. (2020); Ma et al. (2020); Sankar et al. (2020); Wang et al. (2021a); Trivedi et al. (2017); Kumar et al. (2019). Architectures like TGAT Xu et al. (2020a), TGN Rossi et al. (2021), ROLAND You et al. (2022), and others Zhang et al. (2023); Wang et al. (2024); Fan et al. (2021) are designed to model temporal dependencies and evolving structures. Crucially, these works optimize the *model architecture* (how to encode history), whereas GNNUpdater optimizes the *update policy* (when to retrain). These axes are orthogonal: GNNUpdater can serve as a wrapper around any of these backbones (e.g., TGN or ROLAND) to trigger their specific retraining procedures adaptively. We empirically demonstrate this compatibility using TGAT in §5.

**Update Triggering and Drift Detection.** Determining when to retrain models is a key challenge in MLOps, which has led to the development of various monitoring systems Azure (2025); Cloud (2025); Nigenda et al. (2022); NannyML (2023); Evidently (2023) and drift detection methods Gama et al. (2004); Baena-García et al. (2006); Raab et al. (2020); Frias-Blanco et al. (2014). GNNUpdater differs in three aspects: (1) *Graph Focus*: Unlike i.i.d. triggers Smola et al. (2006); Gama et al. (2014); Lu et al. (2018), GNNUpdater detects *hidden drifts* propagated through structural dependencies. (2) *Label-Delay Tolerance*: Instead of requiring immediate feedback Ktena et al. (2019); Frias-Blanco et al. (2014), GNNUpdater handles delayed labels via its performance predictor. (3) *Cost-Awareness*: GNNUpdater frames update timing as an economic decision, balancing performance value against retraining costs via operational knobs Tian et al. (2018); Mallick et al. (2022); Lyu et al. (2024), whereas prior works are cost-agnostic.

# 5 Evaluation

In this section, we evaluate GNNUpdater's adaptive update strategy against several update-trigger baselines. To isolate the effect of the policy itself, **all baselines and GNNUpdater are implemented on top of the same distributed C++ backend** described in §3.4. Detailed evaluations of the system's raw performance against DGL (e.g., a 92.5% reduction in graph operation overhead) are provided in Appendix B. The reported GPU-hour savings therefore reflect only the timing and necessity of updates.

**Datasets and Models.** We evaluate GNNUpdater on three datasets from the Temporal Graph Benchmark (TGB) Huang et al. (2023): `tgbn-genre` Kumar et al. (2019), `tgbn-reddit` Nadiri & Takes (2022), and `tgbn-token` Shamsi et al. (2022). We adopt the node affinity prediction task Huang et al. (2023), which forecasts users' preference scores over all items for the next 7 days. Temporal events are aggregated daily, and predictions are evaluated using the NDCG@10 metric. Due to the task's forecasting horizon, ground-truth labels are available only after a 7-day delay. This delay is characteristic of real-world applications like recommendation systems (where user feedback accrues over time) or fraud detection (where labels require manual investigation). We use three representative GNN models: GraphSAGE Hamilton et al. (2017), GAT Veličković et al. (2018), and TGAT Xu et al. (2020a), all with 2-layer architectures, 100-dimensional embeddings, and sampling 10 neighbors per node per layer. Each is combined with a simple MLP decoder that maps node embeddings to preference scores. Detailed dataset characteristics and model configurations are in Appendix A.1.

**Methodology.** Our experiments simulate a continuous data generation scenario where new data are fed to the system daily for model updating. A base model is initially trained on the first 30% of the total timespan of the dataset. Thereafter, daily aggregated data are sent to GNNUpdater, and the system determines whether to trigger fine-tuning based on the specified update trigger method. The performance predictor's Random Forest regressor is also retrained every batch (daily) using ground-truth labels as they become available (after the 7-day lag) to maintain prediction accuracy. Upon triggering, and in line with common practice Tian et al. (2018); Yuan et al. (2023); Wang et al. (2024), each fine-tuning operation uses the past 365 days of labeled data (offset by 7 days due to the label lag) and is performed for one epoch to balance stability and plasticity. Detailed training configurations are in Appendix A.2.

**Baselines.** We compare GNNUpdater to these baselines (DTP = delayed true performance, PP = predicted performance): (1) **Periodic Update:** Fixed-time interval updates regardless of performance; (2) **PerfDrop-DTP:** Triggers update when 7-day mean NDCG@10 (with 7-day label delay) drops below the predefined threshold $\epsilon$; (3) **PerfDrop-PP:** Triggers update when 7-day mean predicted NDCG@10 drops below the predefined threshold $\epsilon$; (4) **ADWIN-DTP:** ADWIN drift detector Bifet & Gavalda (2007) on DTP stream; (5) **ADWIN-PP:** ADWIN applied to PP stream with identical configuration; (6) **KSWIN-DTP:** Kolmogorov-Smirnov Windowing (KSWIN) drift detector Raab et al. (2020) on DTP stream; (7) **KSWIN-PP:** KSWIN applied to PP stream with identical parameters; (8) **MMD:** Triggers update when the Maximum Mean Discrepancy (MMD) Smola et al. (2006) score exceeds a specified threshold. Detailed method settings and GNNUpdater's sensitivity analysis can be found in Appendix A.3.

## 5.1 Main Results

**Evaluation Metrics.** We use two primary metrics: (1) average downstream task performance (NDCG@10) across streaming batches, and (2) total training GPU hours incurred due to model updates. Importantly, as both GNNUpdater and all baselines utilize the same training engine, the GPU hour metric directly reflects the efficiency of the update *policy*. Our analysis is twofold: first, we compare the average performance achieved by different methods under an identical total training GPU hour constraint; second, we compare the total training GPU hours required by each method to maintain an equivalent level of average performance (defined as an absolute NDCG@10 difference of less than 0.001). All methods were optimized via extensive hyperparameter grid search for each scenario (see Appendix A.4 for tuning strategy and parameters), ensuring GNNUpdater is benchmarked against strong baseline configurations.

**Superior Performance Under a Fixed GPU Hour Budget.** Figure 7a compares average performance when total training GPU hours are equalized across all methods. GNNUpdater improves average NDCG@10 by **5.3% on average** across all datasets and models, with a maximum improvement of **34.0%** (for GAT on `tgbn-token` compared to ADWIN-DTP). The TGAT results in Figure 7a show that the policy also transfers to temporal GNN backbones, not just static ones.

The improvements are most pronounced on `tgbn-token` (up to 34.0%), likely from its complex transaction patterns. Cryptocurrency transaction networks evolve rapidly, causing models to quickly become outdated Shamsi et al. (2022). `tgbn-reddit` shows more moderate gains (up to 2.9%), reflecting its relatively stable social network structure.

While baselines perform competitively on specific datasets (e.g., PerfDrop variants on `tgbn-genre`, periodic updates and MMD on `tgbn-reddit`, KSWIN variants on `tgbn-token`), they consistently trail GNNUpdater. Notably, MMD, despite potential competitiveness, frequently encounters out-of-memory (OOM) errors on large datasets due to its

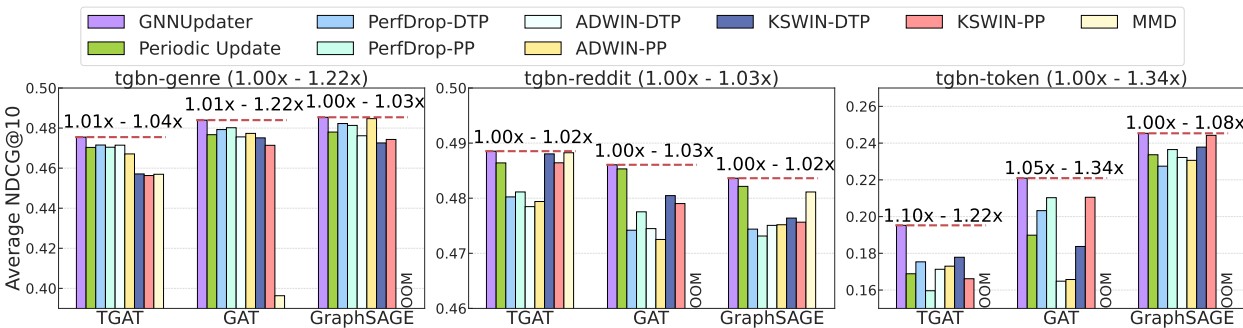

(a) Average performance comparison under equal total training GPU hour constraints across various datasets and models. GNNUpdater improves performance by 5.3% on average.

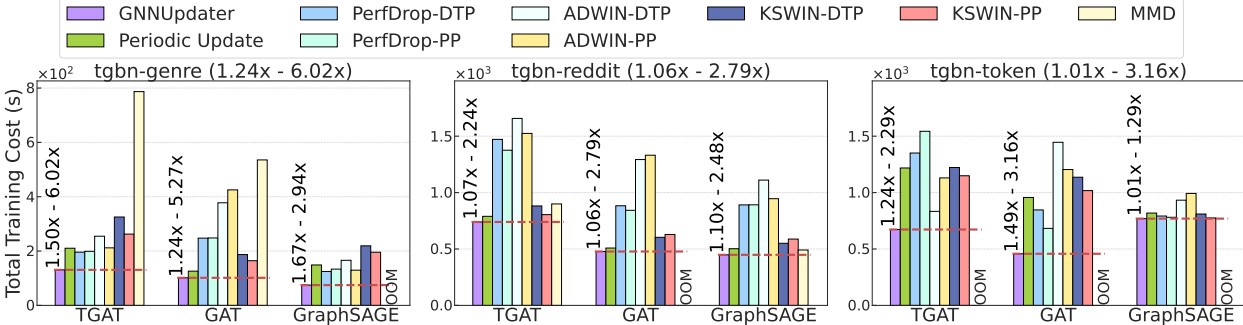

(b) Total training GPU hour comparison when maintaining equivalent average performance (absolute difference < 0.001) across various datasets and models. GNNUpdater reduces training GPU hours by 2.0× on average.

Figure 7: GNNUpdater improves performance at equal training GPU hours or matches their performance with far less training effort. OOM indicates GPU out-of-memory error.

$O(N^2)$ complexity, and fails to capture structural degradation. These results show that the consistent effectiveness of GNNUpdater's update strategy.

**Operational Cost Savings for Equivalent Performance.** Figure 7b shows total training GPU hours required for each method to match the average performance achieved by GNNUpdater. Our approach reduces training GPU hours by a factor of **2.0× on average**, with a maximum reduction of **6.0×** (for TGAT on `tgbn-genre` compared to MMD). On `tgbn-genre`, the best-performing baselines still require 19-40% more training time to reach the same quality. On `tgbn-reddit`, periodic updates remain relatively competitive but still incur 6-9% more overhead than GNNUpdater. For `tgbn-token`, the overhead varies from 0.78% for KSWIN-DTP with GraphSAGE to 33% for PerfDrop-PP with GAT.

## 5.2 Ablation Studies

**Performance Predictor.** We first evaluate the two design choices behind the performance predictor in §3.2: the drift signal used as input, and the regressor used to map that signal to predicted performance.

Table 2 compares candidate embedding difference metrics, with correlations averaged across all datasets and models. Simple node-level metrics like L1/L2 distances show moderate correlation (-0.60) but high variance, indicating they miss important structural changes. Distribution-based metrics like MMD and Central Moment Discrepancy (CMD) Zellinger et al. (2017) are computationally expensive (850ms and 160ms, respectively) while achieving lower or unstable correlations. Our 1-hop neighbor L1 distance achieves both the strongest (-0.80) and most stable (±0.06) correlation while maintaining efficient computation (70ms). While 2-hop neighborhoods marginally improve correlation (-0.81), the increased standard deviation (±0.10) and computation time (240ms) suggest that it introduces noise while being less efficient. These results support the 1-hop neighbor aggregation used in Eq. 3.

Table 2: Comparison of different embedding difference metrics.

| Metric Type | Correlation with NDCG@10 | Compute Time (ms) |
|---|---|---|
| *Node-level Metrics* | | |
| L1 distance | -0.60±0.25 | 20 |
| L2 distance | -0.57±0.25 | 20 |
| *Distribution-based Metrics* | | |
| MMD | -0.40±0.46 | 850 |
| CMD | -0.67±0.30 | 160 |
| *Neighbor-aware Metrics* | | |
| 1-hop Neighbor L1 Distance | **-0.80±0.06** | **70** |
| 2-hop Neighbor L1 Distance | -0.81±0.10 | 240 |

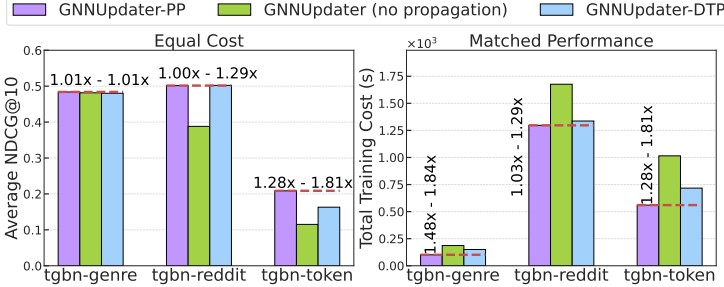

Figure 8: Performance comparison of GNNUpdater variants under equal cost and matched performance scenarios.

Figure 9: Comparison of per-batch overhead (in seconds) of different methods.

Next, Table 3 compares Random Forest (RF) with linear regression (LR) for performance prediction. RF consistently outperforms LR across all datasets and architectures, with error rates as low as 3.51% and correlation up to 0.96. On the complex `tgbn-token` dataset, RF reduces prediction error by over 27% relative to LR. RF still improves over LR on the more stable `tgbn-reddit` dataset. These results show that the feature-to-performance mapping is nonlinear.

**Graph-aware Trigger.** Next, we examine our graph-aware update trigger (§3.3) using GAT by comparing three variants: (1) our full design using predicted performance with label propagation (GNNUpdater-PP), (2) a version that omits the label propagation step and relies solely on performance prediction from the current batch (GNNUpdater (no

Table 3: Performance prediction error and Pearson correlation comparison between Linear Regression (LR) and Random Forest (RF) regressors.

| Dataset | Model | Linear Regression Error(%) | Corr | Random Forest Error(%) | Corr |
|---|---|---|---|---|---|
| `tgbn-genre` | TGAT | 5.93 | 0.60 | **5.14** | **0.69** |
| | GAT | 5.94 | 0.57 | **5.51** | **0.63** |
| | GraphSAGE | 5.66 | 0.41 | **5.12** | **0.52** |
| `tgbn-reddit` | TGAT | 5.21 | 0.91 | **3.97** | **0.94** |
| | GAT | 4.75 | 0.91 | **3.87** | **0.94** |
| | GraphSAGE | 4.65 | 0.90 | **3.51** | **0.93** |
| `tgbn-token` | TGAT | 15.91 | 0.92 | **12.05** | **0.96** |
| | GAT | 18.62 | 0.94 | **13.48** | **0.96** |
| | GraphSAGE | 16.06 | 0.90 | **13.33** | **0.95** |

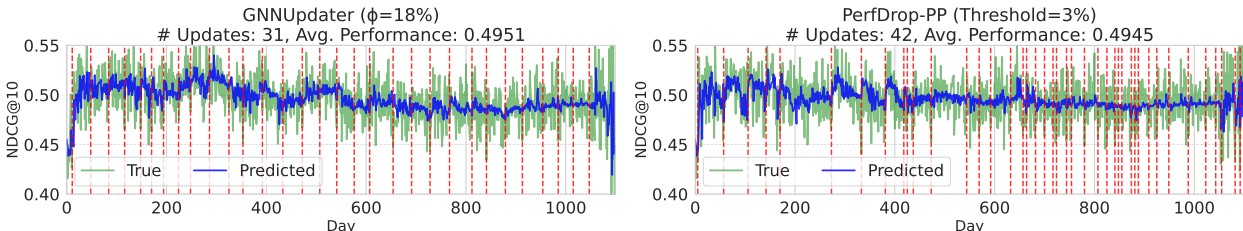

Figure 10: Update trigger patterns comparison between GNNUpdater (left) and PerfDrop-PP (right). Blue/green lines show predicted/true NDCG@10 scores; red dashed lines indicate model updates. GNNUpdater achieves similar performance with fewer, more evenly distributed updates.

propagation)), and (3) a baseline that triggers updates based on delayed true performance (GNNUpdater-DTP). Similar to §5.1, we examine two scenarios, as shown in Figure 8.

Under equal computational budget, our full approach achieves up to 1.81× higher NDCG@10 across datasets. The largest gains are observed on `tgbn-token` compared to variants without label propagation, where complex transaction patterns make hidden drift detection crucial. When targeting matched average performance (absolute difference < 0.001), variants without label propagation require up to 1.84× more training GPU hours. GNNUpdater-DTP, which uses delayed true performance, requires up to 1.48× more training GPU hours despite similar performance. These results show that label propagation and performance prediction together yield better trigger decisions.

**Trigger Update Algorithm's Overhead.** Figure 9 compares the trigger algorithm overhead per batch between GNNUpdater and baselines. We exclude simple methods like periodic updates from this comparison as they incur negligible computational overhead. MMD's reported overhead is measured before encountering out-of-memory errors. Our method incurs comparable computational costs with most baseline methods, requiring about **0.6 seconds** per batch (which includes daily data in our TGB setup).

This low overhead is attributed to three factors: (1) *Embedding Reuse*: GNNUpdater monitors embedding shifts by reusing node embeddings directly from the real-time inference pipeline, avoiding redundant GNN forward passes. (2) *Lightweight Predictor*: The Random Forest regressor is extremely efficient; it is retrained every batch using delayed labels (taking ≈0.25s) and its inference time is negligible. (3) *Efficient LP*: The label propagation step uses the same sparse aggregation operations already optimized in the GNN backend. These optimizations ensure that monitoring remains a small fraction of the total operational cost.

## 5.3 Case Study

To better understand our graph-aware update trigger, we present a detailed case study comparing GNNUpdater with the traditional PerfDrop-PP trigger on the `tgbn-genre` dataset using the GAT model. Figure 10 compares trigger patterns between PerfDrop-PP and GNNUpdater's graph-aware approach. While achieving a similar NDCG@10 (0.4951 vs 0.4945), GNNUpdater requires fewer updates (31 vs 42). GNNUpdater's update trigger demonstrates a more uniform distribution, while PerfDrop-PP shows a clustered pattern. For example, PerfDrop-PP triggers multiple updates in quick succession during days 400-450, 600-800, 800-900, and 1000-1100, yet these clustered updates yield minimal performance improvements. In contrast, GNNUpdater uses graph awareness to assess if performance degradation is widespread and significant before triggering. This structural insight prevents the frequent, less effective updates seen with PerfDrop-PP, leading to more efficient model maintenance with fewer, yet more impactful, updates while preserving performance.

## 6    Conclusion

We propose GNNUpdater, a distributed system for deciding when to update GNN models in continual graph learning. By combining a lightweight performance predictor based on embedding drift with graph-aware label propagation, GNNUpdater improves accuracy by 5.3% on average (up to 34.0%) under fixed GPU-hour budgets and reduces training cost by 2.0× on average at matched accuracy compared to various baselines.

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

# A  Experimental Details

## A.1  Datasets and Models

Table 4: Dataset Statistics. * denotes randomized features. $|d_v|$ and $|d_e|$ show the dimensions of node features and edge features, respectively. The first 30% of the temporal data is used as the initial training period.

|                  | tgbn-genre      | tgbn-reddit    | tgbn-token          |
|------------------|-----------------|----------------|---------------------|
| Domain           | Recommendation  | Social Network | Transaction Network |
| $|V|$            | 1,505           | 11,766         | 61,756              |
| $|E|$            | 17.9M           | 27.2M          | 72.9M               |
| Duration (month) | 52              | 36             | 26                  |
| #Batches         | 1,580           | 1,090          | 785                 |
| $|d_v|$          | 100*            | 100*           | 100                 |
| $|d_e|$          | 1               | 1              | 2                   |

**Testbed.** Unless otherwise specified, we conduct all experiments on two machines, each equipped with 4 x 40 GB A100 GPUs, 256 AMD EPYC 7H12 64-Core processors, 512 GB DRAM, and a 100 Gbps Ethernet network.

**Datasets.** We evaluate GNNUpdater on three real-world datasets from the Temporal Graph Benchmark (TGB) Huang et al. (2023): `tgbn-genre` Kumar et al. (2019), `tgbn-reddit` Nadiri & Takes (2022), and `tgbn-token` Shamsi et al. (2022). Detailed statistics for each dataset are given in Table 4. `tgbn-genre` is a bipartite and weighted interaction network between users and the music genres of songs they listen to Kumar et al. (2019). `tgbn-reddit` is a user and subreddit interaction network Nadiri & Takes (2022). `tgbn-token` is a user and cryptocurrency token transaction network Shamsi et al. (2022).

**Models.** We use three representative GNN models in our experiments: GraphSAGE Hamilton et al. (2017), GAT Veličković et al. (2018), and TGAT Xu et al. (2020a). Following their original implementations, all models use a 2-layer architecture where each layer randomly samples 10 neighbors per node, with 100-dimensional hidden embeddings. For all models, we set dropout to 0.1. For GraphSAGE, we use the mean aggregator. For GAT and TGAT, we use 2 attention heads. For TGAT, the temporal encoding dimension is 100.

## A.2  Training Configurations

For the initial base model training with the first 30% of data, we train each base model for 200 epochs with early stopping, using the Adam optimizer Kingma & Ba (2015) with a learning rate of $1 \times 10^{-3}$. For each fine-tuning operation triggered by any method, we use the past 365 days of labeled data (offset by 7 days due to the inherent label delay in the task) and fine-tune the GNN model for one epoch. The Adam optimizer is used with a fine-tuning learning rate of $1 \times 10^{-4}$. No weight decay is applied during fine-tuning. We use a batch size of 2,000 for `tgbn-genre` and `tgbn-token`, and 4,000 for `tgbn-reddit` due to its larger size.

## A.3  Method Settings and Sensitivity Analysis

For ADWIN and KSWIN, we adopt the `scikit-multiflow` package Montiel et al. (2018) and adjust the default window size to 15 for a fair comparison with other methods. For MMD implementation, we use the `alibi-detect` package Van Looveren et al. (2019) and leverage PyTorch with CUDA acceleration as the backend, since the CPU version is prohibitively slow. We set the performance SLO $\varepsilon$ to 0.50 for both `tgbn-genre` and `tgbn-reddit`, and

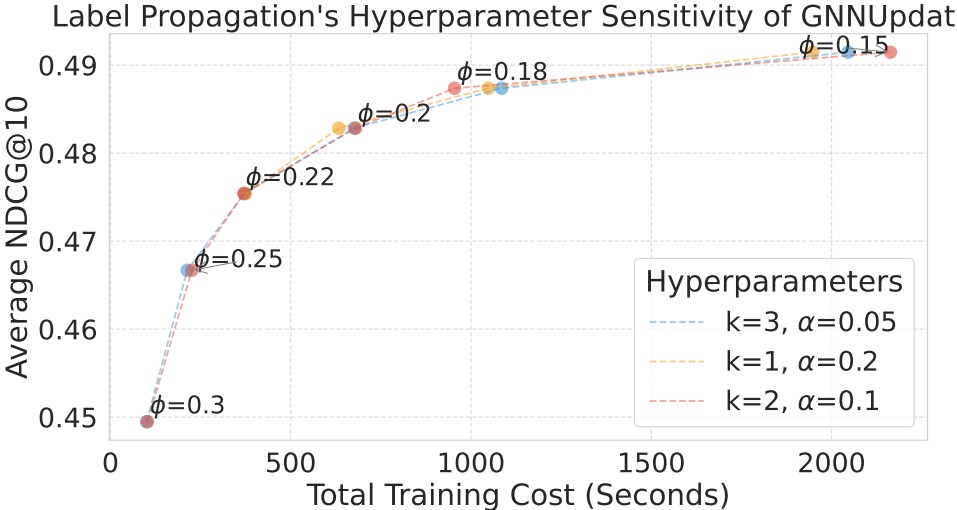

Figure 11: Impact of label propagation hyperparameters on GNNUpdater's performance-cost trade-off, using the TGAT model on the `tgbn-genre` dataset. The close similarity across different settings indicates GNNUpdater's robustness to these specific label propagation hyperparameter choices.

0.25 for `tgbn-token`, based on their initial base model training performance and the recommendations in the original paper Huang et al. (2023).

By default, GNNUpdater combines performance predictor (§3.2) with 2-layer label propagation (§3.3) with propagation layers $k = 2$ and propagation weight $\alpha = 0.1$. We also investigated the sensitivity of GNNUpdater to the choice of its label propagation hyperparameters: the number of propagation layers $k$ and the propagation weight $\alpha$. Figure 11 presents this analysis for the TGAT model on the `tgbn-genre` dataset. Other models and datasets show similar results. It plots the Average NDCG@10 against Total Training Cost for three distinct $(k, \alpha)$ pairs—specifically $(k = 3, \alpha = 0.05)$, $(k = 1, \alpha = 0.2)$, and our default setting of $(k = 2, \alpha = 0.1)$—across a range of problematic node ratio thresholds $(\phi)$. The results demonstrate that all three configurations yield remarkably similar performance-cost trade-off curves. This low sensitivity indicates that GNNUpdater's effectiveness is robust to reasonable variations in these label propagation parameters.

## A.4 Hyperparameter Optimization Strategy

To ensure a rigorous and fair comparison for each dataset and GNN model combination, we performed an extensive grid search over the hyperparameters for GNNUpdater and all baseline methods. For each of the two experimental scenarios described in §5.1, the reported results for every method (including GNNUpdater) correspond to the hyperparameter configuration that optimized its outcome under that specific scenario's objective (i.e., maximizing NDCG@10 for a fixed cost, or minimizing cost for a target NDCG@10). This systematic tuning ensures that GNNUpdater is benchmarked against the most competitive configurations of all baselines. The searched hyperparameters and their respective value ranges for each dataset were as follows:

### A.4.1 `tgbn-genre`

- PerfDrop Variants (%): `[0.005, 0.01, 0.02, 0.03, 0.04, 0.05, 0.07, 0.08, 0.1, 0.15]`

- Periodic Update (Intervals in days): `[14, 30, 60, 90, 180]`

- GNNUpdater (Problematic Node Ratio $\phi$): `[0.15, 0.18, 0.2, 0.22, 0.25, 0.3]`

- ADWIN (Delta $\delta$): `[1.0, 2.0, 5.0, 5.5, 6.0, 6.5, 7.0, 10.0]`

- KSWIN (Significance Level $\alpha_{\text{KSWIN}}$): `[0.001, 0.005, 0.01, 0.05, 0.1, 0.3, 0.5, 0.7, 0.8, 0.9]`

- MMD (Distance Threshold): `[-0.0003, -0.0005, -0.0007, -0.0008, -0.0009, -0.001, -0.0012]`

### A.4.2 `tgbn-reddit`

- PerfDrop Variants (%): `[0.01, 0.02, 0.03, 0.04, 0.05, 0.1]`

- Periodic Update (Intervals in days): `[7, 14, 30, 60, 90]`

- GNNUpdater (Problematic Node Ratio $\phi$): `[0.3, 0.4, 0.42, 0.45, 0.47, 0.5]`

- ADWIN (Delta $\delta$): `[1.0, 2.0, 5.0, 5.5, 6.0, 6.5, 7.0, 10.0]`

- KSWIN (Significance Level $\alpha_{KSWIN}$): `[0.001, 0.005, 0.01, 0.05, 0.1, 0.3, 0.5, 0.7, 0.8, 0.9]`

- MMD (Distance Threshold): `[0.0015, 0.0012, 0.001, 0.0007, 0.0005, 0.0003]`

### A.4.3 `tgbn-token`

- PerfDrop Variants (%): `[0.1, 0.2, 0.3, 0.4, 0.5]`

- Periodic Update (Intervals in days): `[7, 14, 30, 60, 90]`

- GNNUpdater (Problematic Node Ratio $\phi$): `[0.3, 0.4, 0.45, 0.5, 0.55, 0.56, 0.57, 0.58, 0.6, 0.7]`

- ADWIN (Delta $\delta$): `[1.0, 2.0, 5.0, 5.5, 6.0, 6.5, 7.0, 10.0]`

- KSWIN (Significance Level $\alpha_{KSWIN}$): `[0.008, 0.01, 0.05, 0.1, 0.3, 0.5, 0.7, 0.8, 0.9]`

- MMD (Distance Threshold): `[0.2, 0.15, 0.13, 0.1, 0.08, 0.07, 0.05]`

Table 5: Graph operation overhead with DGL across all datasets (in milliseconds per batch)

| Dataset | Component | GNNUpdater | DGL |
|---|---|---|---|
| `tgbn-genre` | Graph Updates | **20.88** | 118.58 |
| | Neighbor Finding | **2.65** | 3.20 |
| | Feature Loading | **1.76** | 214.55 |
| | Total | **25.29** | 336.33 |
| `tgbn-reddit` | Graph Updates | **52.32** | 130.26 |
| | Neighbor Finding | **10.74** | 12.31 |
| | Feature Loading | **10.70** | 235.30 |
| | Total | **75.76** | 377.87 |
| `tgbn-token` | Graph Updates | **9.09** | 81.03 |
| | Neighbor Finding | 4.48 | **3.07** |
| | Feature Loading | **4.49** | 103.07 |
| | Total | **18.06** | 187.17 |

## B  System Efficiency and Scalability

**Graph Operation Cost Analysis.** We analyze the graph operation cost across all datasets. Table 5 compares the graph operation overhead between GNNUpdater and DGL (v2.4.0). Both systems employ UVA-based neighbor finding and GPU feature caching (caching features for 1e4 nodes and 1e7 edges). Our system achieves efficient graph operations through three key optimizations: (1) Block-based streaming graph storage reduces graph update time by

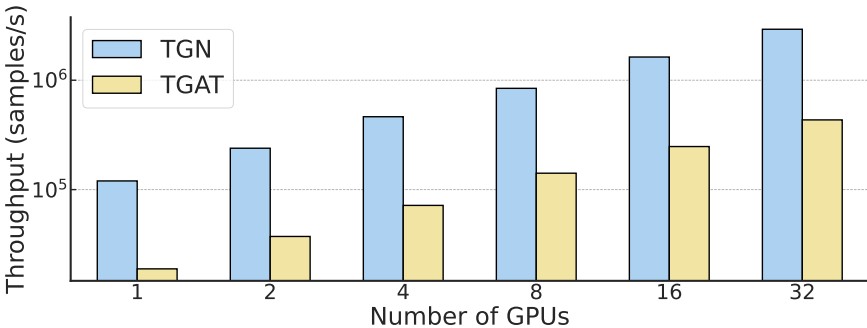

Figure 12: Training throughput scaling with different numbers of GPUs on the GDELT dataset. GNNUpdater maintains near-linear scaling within single machine (1-8 GPUs) and high scaling efficiency (>70%) across multiple machines.

82-88% across datasets by eliminating the need for full graph reconstruction; (2) Our GPU-based neighbor finder leverages 22.24 MB of metadata stored on the GPU, achieving comparable or better performance despite using a flexible linked list data structure rather than DGL's static, compact Compress Sparse Row (CSR) representation; and (3) Optimized cache management for feature loading cuts access time by 95-99% across datasets. Together, these enhancements yield substantial reductions in graph operation overhead compared to DGL: 92.5% for `tgbn-genre`, 80.0% for `tgbn-reddit`, and 90.4% for `tgbn-token`.

Table 6: Multi-machine training performance comparison using 32 GPUs. GNNUpdater achieves faster graph construction and higher training throughput than DGL.

| Metric | DGL | GNNUpdater | Improv. |
|---|---|---|---|
| Graph Building Time (s) | 7778 | 5246 | 1.32× |
| **Throughput (samples/s)** | | | |
| GraphSAGE | 124.1k | 181.6k | 1.46× |
| GAT | 146.9k | 193.7k | 1.32× |

**Multi-GPU Scaling.** We evaluate GNNUpdater's scalability on two additional large-scale datasets: GDELT Zhou et al. (2022), a global event interaction network (17K nodes, 191M edges, 33 GB of edge features), and MAG Zhou et al. (2022), an academic graph (122M nodes, 1.3B edges, 175 GB of node features). All experiments are conducted on Amazon EC2 g4dn.metal instances, each equipped with 8 T4 GPUs. Using GDELT—which fits in single-machine memory—we evaluate the training throughput of TGN Rossi et al. (2021) and TGAT Xu et al. (2020a) with varying numbers of GPUs from 1 to 32. As shown in Figure 12, GNNUpdater achieves near-linear speedup on a single machine and maintains 71.9% (TGN) and 76.2% (TGAT) scaling efficiency of the ideal linear-scaling performance on 32 GPUs.

**Distributed Training.** For graphs exceeding single-machine capacity, we evaluate distributed training on partitioned MAG graph using 32 GPUs. As shown in Table 6, GNNUpdater achieves up to 1.46× training speedup compared to DGL while reducing METIS graph partitioning Karypis & Kumar (1998) and construction time by 1.32×. We further validate scalability on a synthetic LDBC graph (5.1B edges) Angles et al. (2020) distributed across 8 machines with TGN - GNNUpdater maintains efficient training with 556k samples/second throughput. These results demonstrate our system optimizations enable effective scaling from single GPU to large distributed deployments.

