# OpenReview forum: "GNNUpdater: Adaptive Self-Triggered Training Framework on Dynamic Graphs"
_TMLR — Under review for TMLR_

### Review · Reviewer_qYar · 2026-06-18

**Summary Of Contributions:**

The paper proposes GNNUpdater, an adaptive framework for deciding when to retrain GNNs on dynamic graphs. According to the paper, periodic retraining can waste GPU budget, while reactive retraining based on observed performance can arrive too late because labels are delayed. The paper argues that dynamic graphs introduce an additional challenge, “hidden drift,” where local structural changes can affect multi-hop neighborhoods and degrade predictions on nodes that are not directly active.

The main technical contribution is a two-part trigger mechanism. First, the paper uses embedding shift between reference embeddings after the last update and current embeddings during inference to predict model performance before ground-truth labels arrive. It aggregates drift over a node and its 1-hop neighbors, then trains a lightweight random-forest regressor to map drift-related features to predicted NDCG@10. Second, it proposes a graph-aware update trigger that marks active nodes as problematic when predicted performance falls below an SLO, diffuses these labels through the graph using label propagation, and retrains only when the estimated global degradation ratio exceeds a threshold. The paper also presents a system-level contribution: a distributed streaming-GNN library with block-based dynamic graph storage, GPU-based neighbor sampling, feature caching, and distributed training support.

Key strengths include the practical framing of retraining as a cost–accuracy trade-off, the focus on delayed labels and graph-structured drift, and the attempt to evaluate both algorithmic trigger quality and systems overhead. The paper is also clear about operational knobs such as the SLO threshold and the global degradation threshold.

Key weaknesses are that the novelty of some components may be incremental, since embedding drift, performance prediction, and label propagation are standard tools combined for this setting. The evaluation is somewhat narrow: all main task results use TGB node-affinity prediction with NDCG@10 and a fixed seven-day label delay, so it is unclear how well the approach generalizes to other dynamic-graph tasks, stronger temporal GNNs, different retraining protocols, or more realistic cost models. I have concerns about whether the performance predictor is fairly usable in deployment, since it is retrained daily using delayed labels and tuned extensively per dataset, and whether the “hidden drift” claim is directly validated beyond aggregate ablations.

**Audience:**

Yes

**Audience Explanation:**

At least some members of the TMLR audience would likely be interested in the paper’s findings. The paper addresses a practical and underexplored question in dynamic graph learning: when to retrain a deployed GNN under delayed labels and limited compute budget. This is relevant to researchers and practitioners working on temporal graphs, continual graph learning, MLOps for graph models, recommendation systems, fraud detection, and large-scale GNN infrastructure. The findings in the paper are potentially useful because they suggest that retraining decisions can be improved by using embedding drift as a proxy for delayed performance and by propagating degradation signals over the graph structure. Even if some of the evidence is not fully convincing, the problem formulation and empirical comparisons may inform future work on cost-aware dynamic GNN maintenance.

**Broader Impact Concerns:**

There is no broader impact statement in the paper.

The main broader-impact issue is that the motivating applications include recommendation, fraud detection, financial transaction monitoring, and credit/default prediction. In these settings, adaptive retraining policies can affect high-stakes decisions about users, customers, or accounts. A system that delays retraining to save compute may prolong degraded or biased predictions for some subpopulations, while a trigger based on graph propagation may amplify localized errors or flag structurally connected groups as “problematic.” This could have fairness implications, especially in graphs with homophily or demographic/community structure. The paper should also discuss operational risks. Because GNNUpdater relies on embedding drift and delayed labels, it may fail under semantic drift, adversarial graph manipulation, or label-quality issues. In fraud or financial applications, adversaries might attempt to manipulate graph structure or activity patterns to avoid triggering updates, or conversely to cause unnecessary retraining. The compute-saving framing should be balanced against reliability and safety requirements in high-stakes deployments.

**Claims And Evidence:**

No

**Claims Explanation:**

The paper provides several pieces of evidence supporting its main empirical claims, including comparisons against periodic, performance-drop and MMD baselines; ablations for the embedding-drift predictor and label-propagation trigger; and system-efficiency measurements against DGL. These results are generally clear and suggest that the proposed trigger can reduce unnecessary retraining while maintaining or improving performance on the evaluated TGB datasets.

However, I do not think the evidence is fully convincing for the breadth of the claims. The paper makes strong claims about delayed labels, hidden drift, billion-edge dynamic graphs, and broad applicability to deployed GNN systems, but the main algorithmic evaluation is limited to three TGB node-affinity datasets with the same metric, a fixed 7-day label delay, and a particular retraining protocol. The “hidden drift” motivation is plausible, but it is not directly validated with a controlled experiment showing that inactive/multi-hop regions degrade and that label propagation specifically detects them. The performance predictor is also trained and tuned with delayed labels and extensive per-dataset hyperparameter search, which raises questions about how robust it would be under deployment shift or different SLO/cost settings. Finally, some claims about cost-awareness are only indirectly supported by threshold sweeps and GPU-hour counts, rather than by a realistic economic objective or application-level cost model. Overall, the evidence is promising and mostly clear, but I would not say the submission’s claims are fully supported in an accurate and convincing way.

**Requested Changes:**

- The paper motivates the graph-aware trigger by arguing that local changes can silently degrade predictions in inactive multi-hop neighborhoods, but the experiments do not isolate this phenomenon clearly. Please add a controlled analysis showing that degradation propagates to inactive or multi-hop nodes, and that label propagation detects these failures earlier or more accurately than active-node-only monitoring.
- The predictor is retrained daily using delayed labels and is tuned per dataset. Please explain what historical labeled data is required before deployment, how much delayed feedback is needed for stable predictor training, and how the method behaves during cold start or under major semantic drift.
- The paper frames retraining as an economic optimization, but the evaluation mostly reports GPU time and threshold sweeps. Please connect the trigger threshold more explicitly to the objective in Eq. 1, or evaluate under realistic cost/utility settings where prediction loss and retraining cost are jointly measured.
- The main experiments use a fixed 7-day delay and a 365-day fine-tuning window. Since delayed labels are central to the paper, please evaluate different label delays and retraining windows, or justify why the reported setting is representative.
- The main algorithmic evidence is limited to TGB node-affinity prediction with NDCG@10. Please either add experiments on additional dynamic-graph tasks, such as node classification, link prediction, or fraud/anomaly detection, or temper claims about general dynamic-graph/GNN deployment.
- The paper states that all methods are extensively tuned, but it is not always clear how equal-budget and matched-performance settings are constructed. Please provide enough detail to reproduce the selection of hyperparameters for each baseline, how failed/OOM configurations are handled, and whether validation data or test data influenced tuning.
- Many reported gains are small on some datasets, especially tgbn-reddit. Please report variance over multiple seeds or temporal splits, and indicate which improvements are statistically meaningful.
- The paper includes both an adaptive update policy and a custom distributed system. Although the authors state that baselines use the same backend in the main experiments, the narrative sometimes blends policy benefits with systems benefits. A clearer separation would help readers understand which claims come from the trigger algorithm and which come from implementation optimizations.
- Please add a more explicit limitations section covering cases where embedding drift may not correlate with performance, where label propagation may over-smooth or amplify noise, and where the graph structure is too sparse, too dense, or adversarially changing for the trigger to be reliable.

---

### Review · Reviewer_Eqah · 2026-06-25

**Summary Of Contributions:**

The paper proposes Adapting Graph Neural Networks (GNNs) to evolving, dynamic graphs at scale is operationally challenging and expensive. Retraining models too frequently wastes costly GPU resources, while waiting too long leads to performance degradation. Determining the optimal time to trigger an update is further complicated by two factors: label delay (ground truth labels arrive much later than predictions) and hidden drift (structural dependencies cause changes to propagate across multiple graph hops, unexpectedly harming performance). The authors introduce GNNUpdater, an adaptive framework designed to decide exactly when to trigger GNN retraining. It consists of two key components i) A performance predictor that estimates model quality by measuring shifts in node embeddings, bypassing the need for immediate ground-truth labels. ii) A graph-aware update trigger that leverages label propagation to detect when performance degradation has spread widely across the graph structure

**Additional Comments:**

No comment

**Audience:**

No

**Audience Explanation:**

There are two primary areas where the manuscript requires further refinement to strengthen its contribution:

Lack of Theoretical Justification: The current work relies heavily on empirical validation but lacks a rigorous theoretical or mathematical framework to formally justify the proposed methodology. Providing a theoretical grounding for the framework’s underlying mechanics would significantly bolster the validity of the approach.

Methodological Novelty and Generalizability: The reliance on Personalized PageRank for the Graph-Aware Update Trigger raises concerns regarding the novelty of the technical contribution, as this is a well-established method in graph literature. Furthermore, because this technique is fundamentally designed for homophilic structures (where connected nodes share similar characteristics), the current formulation may not generalize effectively to heterophilic datasets. The authors should address how the framework might be adapted to maintain robust performance across varying graph typologies.

**Broader Impact Concerns:**

The generalizability and reliability of the impact are restricted by two core factors, without a robust theoretical framework or mathematical guarantees underpinning the update trigger, deploying this system in mission-critical applications. Also, by relying fundamentally on Label Propagation based on random walk, the system's benefits are disproportionately skewed toward homophilic graph systems

**Claims And Evidence:**

No

**Claims Explanation:**

The paper does not provide source code, the link is expired.

**Requested Changes:**

Based on the evaluation of the manuscript, I recommend a decision of Reject. The limitations identified regarding theoretical justification and methodological novelty are fundamental to the core contribution, and addressing them would require a substantial restructuring and rewriting of the work that falls outside the scope of standard revisions.

---

### Review · Reviewer_xKif · 2026-07-20

**Summary Of Contributions:**

This paper proposes GNNUpdater, a framework for deciding when to retrain GNNs on dynamic graphs. The method targets two practical issues: delayed labels and graph changes that may affect nodes beyond those observed in the latest batch. It combines an embedding-shift-based performance predictor with label propagation to estimate graph-wide degradation, and triggers retraining when the estimated fraction of problematic nodes exceeds a threshold.

The paper evaluates the method on three Temporal Graph Benchmark datasets with GraphSAGE, GAT, and TGAT. It also presents a distributed implementation with streaming graph storage, GPU sampling, caching, and multi-GPU training. The reported results suggest that GNNUpdater can improve performance under a fixed training budget or reduce retraining cost at similar performance.

**Audience:**

Yes

**Audience Explanation:**

The question of when to retrain a model is practically important and less studied than how to update a dynamic GNN. Researchers working on temporal graphs, continual learning, drift detection, recommender systems, and ML systems would likely find the problem and results relevant.

In particular, the finding that neighborhood-level embedding shifts can help predict future model degradation is interesting, even if the final method is relatively simple. The system implementation and scaling experiments may also be useful to graph-systems researchers.

**Claims And Evidence:**

No

**Claims Explanation:**

My biggest concern is the evaluation protocol. The paper states that hyperparameters are selected separately to maximize performance under a fixed budget or minimize cost at matched performance, but it is unclear whether this tuning is done on a separate temporal validation period. If thresholds are selected using the same stream on which results are reported, the comparison may be optimistic. The paper should clearly separate chronological validation and test periods.

The paper also reports mostly single-run results without error bars or confidence intervals. Some gains, especially on tgbn-reddit, are quite small, so it is difficult to judge whether they are reliable.

The ablation shows that label propagation improves the final cost-performance trade-off, but it does not directly demonstrate the claimed “hidden drift” mechanism. It would be useful to show whether nodes identified only through propagation actually have worse future performance, and whether the method detects this degradation earlier than non-graph-aware triggers.

The connection between the proposed trigger and the economic objective is also somewhat loose. The objective includes an explicit trade-off parameter between loss and training cost, while the implemented method uses a threshold selected by grid search. I would describe the method as cost-aware rather than claiming that it directly optimizes the stated objective.

**Requested Changes:**

1. Use a chronological validation period for selecting all trigger thresholds and baseline hyperparameters, followed by a held-out test period.

2. Report results over multiple random seeds with standard deviations or confidence intervals, and include complete numerical tables.

3. Directly evaluate hidden-drift detection, for example by measuring the accuracy and lead time of propagated degradation signals on inactive or multi-hop nodes.